# Age, Origin and Functional Study of the Prevalent LDLR Mutation Causing Familial Hypercholesterolaemia in Gran Canaria

**DOI:** 10.3390/ijms241411319

**Published:** 2023-07-11

**Authors:** Nicolás M. Suárez, Shifa Jebari-Benslaiman, Roberto Jiménez-Monzón, Asier Benito-Vicente, Yeray Brito-Casillas, Laida Garcés, Ana M. González-Lleo, Antonio Tugores, Mauro Boronat, César Martin, Ana M. Wägner, Rosa M. Sánchez-Hernández

**Affiliations:** 1Instituto Universitario de Investigaciones Biomédicas y Sanitarias, Universidad de Las Palmas de Gran Canaria, 35016 Las Palmas de Gran Canaria, Spain; nicolas.martel@ulpgc.es (N.M.S.); robertojm87@gmail.com (R.J.-M.); yeray.brito@ulpgc.es (Y.B.-C.); agonzalezlleo@gmail.com (A.M.G.-L.); mborcor@gmail.com (M.B.); 2Departamento de Bioquímica y Biología Molecular, Instituto Biofisika (UPV/EHU, CSIC), Universidad del País Vasco UPV/EHU, Bilbao, 48940 Leioa, Spain; shifa.jebari@ehu.eus (S.J.-B.); asier.benito@ehu.eus (A.B.-V.); laidagaal@gmail.com (L.G.); cesar.martin@ehu.eus (C.M.); 3Sección de Endocrinología y Nutrición, Complejo Hospitalario Universitario Insular Materno-Infantil de Gran Canaria (CHUIMI), 35016 Las Palmas de Gran Canaria, Spain; 4Unidad de Investigación, CHUIMI, 35016 Las Palmas de Gran Canaria, Spain; uichuimi@gmail.com

**Keywords:** familial hypercholesterolaemia, founder effect, mutation, *LDLR* gene, origin, Gran Canaria

## Abstract

The p.(Tyr400_Phe402del) mutation in the LDL receptor (*LDLR*) gene is the most frequent cause of familial hypercholesterolaemia (FH) in Gran Canaria. The aim of this study was to determine the age and origin of this prevalent founder mutation and to explore its functional consequences. For this purpose, we obtained the haplotypic information of 14 microsatellite loci surrounding the mutation in one homozygous individual and 11 unrelated heterozygous family trios. Eight different mutation carrier haplotypes were identified, which were estimated to originate from a common ancestral haplotype 387 (110–1572) years ago. This estimation suggests that this mutation happened after the Spanish colonisation of the Canary Islands, which took place during the fifteenth century. Comprehensive functional studies of this mutation showed that the expressed LDL receptor was retained in the endoplasmic reticulum, preventing its migration to the cell surface, thus allowing us to classify this *LDLR* mutation as a class 2a, defective, pathogenic variant.

## 1. Introduction

Familial hypercholesterolaemia (FH, OMIM 144400) is an autosomal codominant disorder that affects 34 million people worldwide [1]. FH is characterised by increased low-density lipoprotein cholesterol (LDL-C) concentrations, which lead to premature atherosclerotic cardiovascular disease (ASCVD) and cholesterol deposits in the cornea and tendons [2].

FH is caused by an array of pathogenic variants affecting genes that regulate cholesterol metabolism [3]. Most of these pathogenic variants are located in the LDL receptor (*LDLR*) gene, resulting in 80% of the cases of FH, with more than 4000 variants described so far in the Human Gene Mutation Database. The heterozygous form of FH (HeFH) is the most common, with a prevalence of 1:200–250 people [4], whereas the more severe homozygous form (HoFH) occurs with a frequency of 1:250,000 to 360,000 [5].

The genetic isolation of certain populations has led to an increase in the frequencies of some variants via founder effects. This phenomenon has been reported in Afrikaners [6], Ashkenazi Jews [7], French Canadians [8], Lebanese [9], Finns [10], and recently in the Canary Islands population [11]. Although FH displays broad genetic heterogeneity in general—the most frequent variant in Spain represents only 7% of FH cases [12]—almost 70% of FH cases with a positive genetic diagnosis in the island of Gran Canaria are due to a single mutation (p.[Tyr400_Phe402del]) in *LDLR*, which is associated with severe hypercholesterolaemia and increased cardiovascular risk [13]. Interestingly, this mutation has only been reported in Gran Canarians [14], suggesting it originated in the population of this island.

Although the genetic background of Canarian people is mainly Caucasian as a result of the Spanish conquest during the 15th century [15], several studies have demonstrated that a significant aboriginal contribution from North African populations still remains [16,17]. This aboriginal contribution has been linked to the spread of inheritable disorders, as demonstrated by the presence of specific mutations causing rare, recessive disorders, such as Wilson’s disease [18] and type 2 tyrosinemia [19].

Estimating the age of the p.(Tyr400_Phe402del) mutation would be instrumental to determine whether this genetic variant arose in Gran Canaria or has been introduced by contemporary migration. In addition, it can provide relevant information about the evolutionary processes driving its current frequency in the population of Gran Canaria. Several methods using genotyping data have been developed in order to estimate the age of the variant of interest or the most recent common ancestor of its carriers [20,21,22].

Furthermore, although family studies have been performed that confirm the segregation of this variant with hypercholesterolaemia and ASCVD [11], an in vitro functional study to confirm pathogenicity has not yet been performed. 

The aim of this study is to estimate the age of this founder mutation to understand the genetic epidemiology of this variant in the population of Gran Canaria and to perform a functional study in order to propose a mechanism by which the p.(Tyr400_Phe402del) mutation generates the FH phenotype.

## 2. Results

### 2.1. Demographic, Clinical and Genetic Characterisation

A total of 11 unrelated family trios of p.(Tyr400_Phe402del)-mutation carriers and a homozygous individual were analysed. The clinical information of mutation carriers is described in Table 1. Briefly, p.(Tyr400_Phe402del) carriers exhibit very high levels of LDL-c, and over 42% of them present tendinous xanthomas. In addition, the Dutch Lipid Clinic Network (DLCN) score is higher than 8 in all the subjects.

Fourteen autosomal microsatellite loci flanking the p.(Tyr400_Phe402del) mutation were analysed for all 82 individuals (34 mutation carriers and relatives, and 48 controls). All but one locus (trinucleotide) contained dinucleotide repeats and presented from 6 to 16 different alleles (Table 2). Significant deviations from HWE due to heterozygote deficiency were detected in two loci (L5 and R1). As expected, considering their genomic proximity, several locus combinations showed significant LD (Appendix A).

### 2.2. Age of the p.(Tyr400_Phe402del) Mutation

To estimate the age of the p.(Tyr400_Phe402del) mutation, haplotypic information for the 14 microsatellites analysed were deduced in carriers using their first-degree relatives. A total of eight different haplotypes were identified (Table 3).

Assuming a ‘correlated’ genealogy, which considers the possibility of the mutation age being more recent than the most recent common ancestor for the analysed population, the mutation arose 15.5 generations ago, with a confidence interval of 4.4–62.9.

**Table 3 ijms-24-11319-t003:** Haplotypes (constellation of alleles (size in base pairs) for the fourteen microsatellites analysed) carrying the p.(Tyr400_Phe402del) mutation, and estimation of its age in generations (and years, assuming 25 years per generation) with a 95% confidence interval (CI).

		Microsatellite Markers (Distance (cM) from the Mutation)														
		L10 (4.69)	L9 (4.19)	L8 (4.04)	L7 (3.82)	L6 (3.20)	L5 (2.89)	L4 (1.85)	L3 (0.38)	L2 (0.05)	L1 (0.01)	p.(Tyr400_Phe402del)	R1 (0.99)	R2 (1.31)	R3 (2.16)	R4 (4.17)
Haplotypes (frequency)	1 (54.8%)	327	193	241	225	245	225	226	143	163	119		374	159	256	205
	2 (3.2%)	329	193	243	225	245	225	226	143	163	119		374	159	256	205
	3 (16.1%)	327	193	241	225	245	225	226	143	163	119		374	159	260	205
	4 (12.9%)	335	203	241	225	245	225	226	143	163	119		374	159	256	205
	5 (3.2%)	335	203	241	225	245	225	226	143	163	119		374	159	256	207
	6 (3.2%)	327	193	241	225	245	225	224	143	163	119		374	159	260	205
	7 (3.2%)	335	193	241	225	245	225	226	143	163	119		374	159	256	205
	8 (3.2%)	339	203	241	239	245	225	226	143	163	119		388	159	260	205
		Generations (years)						CI-lower					CI-upper			
	Assuming a ‘correlated’ genealogy	15.5 [387]						4.4 [110]					62.9 (1572)			

Cells in orange indicate the location of the mutation (see Figure 1, Panel B). Cells in blue indicate the different alleles that define a specific haplotype. Complete genotyping information is described in Appendix A.

**Figure 1 ijms-24-11319-f001:**
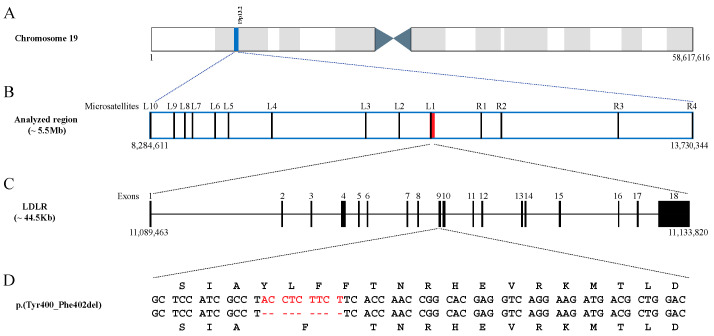
Schematic representation of (i) the genomic location of the analysed region (in blue) within chromosome 19 (**A**); (ii) the genomic location of the analysed microsatellites (L, left side of the mutation; R, right side of the mutation) and the *LDLR* gene (in red) within the analysed region (**B**); (iii) the structure of the *LDLR* gene (**C**); and (iv) the p.(Tyr400_Phe402del) mutation (in red), with the corresponding amino acid changes (top and bottom sequences) (**D**). Numbers below panels A–C indicate the location in base pairs corresponding to the Human Genome Assembly (GRCh38.p13).

### 2.3. Expression of the p.(Tyr400_Phe402del) LDLR Variant in CHO-ldlA7 Cells

Expression of the p.(Tyr400_Phe402del) LDLR variant was analysed by Western blot and flow cytometry in CHO-*ldl*A7-transfected cells, as described in Section 4 (Materials and Methods). For surface expression analysis by flow cytometry, two variants were used as internal method controls, p.(Trp87)* (a null allele mutant) and the Ex3_4del LDLR variant that is expressed to a similar extent as wt LDLR but is a class 3 variant with 100% impaired binding activity [23]. As shown in Figure 2A, the p.(Tyr400_Phe402del) LDLr variant is not expressed at the membrane surface compared to wt-transfected cells (wt: 100 ± 2.0; p.[Tyr400Phe402del]: 14.8 ± 4.5).

To confirm whether p.(Tyr400_Phe402del) is not expressed in its mature form, LDLR expression was assessed 48 h post-transfection. As shown in Figure 2B, only the expression of immature p.(Tyr400_Phe402del) was detected by Western blot, confirming the flow cytometry results.

### 2.4. LDL Uptake Activity of the p.(Tyr400_Phe402del) LDLR Variant in CHO-ldlA7 Cells

Activity of the p.(Tyr400 Phe402del) LDLR variant was also assessed in CHO-*ldlA*7-transfected cells as described in Section 4 (Materials and Methods). As shown in Figure 3, LDL uptake by p.(Tyr400 Phe402del) showed only residual LDLR activity (wt: 100 ± 3; p.[Tyr400 Phe402del]: 17 ± 5).

### 2.5. p.(Tyr400_Phe402del) LDLR Variant Classification by Confocal Microscopy 

To further analyse the type of defect produced by the in-frame deletion of Tyr400_Phe402 residues, we studied whether the immature expressed form of the p.(Tyr400_Phe402del) LDLR variant colocalised with calregulin, an endoplasmic reticulum (ER) marker, using a confocal microscope. Confocal images show that the variant is expressed in transfected cells, but remains clearly retained in the ER, as indicated by the high colocalisation with calregulin (Figure 4), which corroborates the experimental data obtained by flow cytometry and Western blot. Accordingly, the p.(Tyr400_Phe402del) *LDLR* variant should be classified as a class 2a, defective, pathogenic variant.

## 3. Discussion

In this study we aimed to reveal the age and origin of the p.(Tyr400_Phe402del) *LDLR* mutation and the functional consequences on the expressed LDLR variant. To this end, we selected an optimally distanced [24] set of 14 microsatellites spanning 8.86 cM around the variant (p.[Tyr400_Phe402del]), and applied a method based on ancestral segment lengths [22] using fine mapping with LD. In the analysed cohort, we identified eight different haplotypes. Considering a recurrent 9 bp deletion is extremely rare, we assumed the different variant carrier haplotypes detected in the Gran Canaria population derive from a common ancestral haplotype (i.e., correlated genealogy). Therefore, this scenario fits with a genetic signature of a founder effect, in which all the mutation carriers have inherited the variant from a common ancestor arising in the population about 387 years ago. Although the confidence interval obtained was rather wide (110 to 1572 years), this estimation postdates the one-century-long Spanish colonisation of the Canary Islands, which ended in 1496 with the surrender of Tenerife [15]. After this dramatic episode, the European colonisation of the Canary Islands involved a mix of Spanish, Portuguese, Italian and Flemish colonisers, who, in addition to the sub-Saharan Africans and Moorish slaves’ contribution [25], have provided the genetic background of the contemporary Canarian population.

The most plausible scenarios supporting the high frequency of this variant in the Gran Canarian population is either the result of gene flow from any of the postcolonisation sources, or an isolated mutational event in the settled Gran Canarian population. Although gene flow has been proposed as the evolutionary process of introducing a different *LDLR* mutation (G197del) in Israel and Lithuania [26], in the case of Gran Canaria, several facts point to a mutational event in the population inhabiting the island after the Spanish colonisation: (i) the first reference to this mutation in the literature refers to participants from a local hospital (Hospital Universitario Dr. Negrín de Gran Canaria) [14]; (ii) the mutation has not been found in mainland Spain nor elsewhere; and (iii) the genetic characteristics and the geographic isolation of the population have been previously shown to facilitate the expansion of genetic variants, causing both recessive [18,19] and dominant disorders [27].

An additional evolutionary process that can facilitate the predominance of specific variants in a population is positive selection. Although a heterozygote advantage has been proposed in other disease-associated variants [28] and HeFH is the most common form of FH in Gran Canaria, this mechanism does not seem to have influenced the current incidence of the variant (p.[Tyr400_Phe402del]). Indeed, carriers of this variant present a higher than expected prevalence of type 2 diabetes [11], as opposed to the view of FH being protective against this disease [29,30]. In addition, as we demonstrate in this study, the p.(Tyr400Phe402del) *LDLR* variant leads to a defective protein. Specifically, the in-frame deletion occurring in the p.(Tyr400Phe402del) *LDLR* variant causes the removal of a tyrosine residue from a highly conserved motif in the first YWTD domain of the LDLR polypeptide. This constitutes one of the six four-stranded beta-sheets (“blades”) that maintain the domain structure, which is determinant for the correct folding of the β-propeller domain [31]. As a result, this in-frame deletion of three residues may trigger the “quality control” machinery of the ER that blocks the trafficking of misfolded proteins [32], thus preventing the migration of the expressed protein to the cell surface and leading to a very severe FH phenotype. Consequently, we can classify the p.(Tyr400Phe402del) *LDLR* mutation as a class 2a, defective, pathogenic *LDLR* variant.

Considering the high prevalence of this class 2a *LDLR* variant in the population of Gran Canaria, the establishment of a rapid diagnostic test to screen the population for the presence of this particular variant is paramount. This will clearly assist clinicians in the diagnosis of this important disease and will allow for the initiation of timely therapeutic interventions. Indeed, this population-based diagnostic strategy is the current routine, not only at our centre, which provides assistance to the Southern and Eastern regions of the island, but also in the other main hospital of Gran Canaria, thus providing full coverage for the island population.

We acknowledge that our study has some limitations. First, the geographic region of the cohort is restricted. However, the sample size surpasses that of other studies on dominant diseases. In addition, unrelated variant-carriers were selected, in order to maximise the representation of the population affected with HF in Gran Canaria. Second, we opted for a genotype-based method, which cannot assure the sequence of the analysed region is identical among subjects sharing the haplotypes identified in this study. However, this method has been widely applied in other studies dating mutations. Furthermore, the microsatellite markers were carefully selected to be optimally distanced and informative, as demonstrated by the identification of recombination points at both sides of the mutation. Third, the methodology applied may have underestimated the age of the variant under investigation, an artefact that is more evident in growing populations [33]. In this regard, we are currently conducting whole genome sequencing in a selected group of variant-carriers, which will not only help us corroborate or refine our estimation but also will provide an opportunity to identify potential modifier genes that may explain the phenotypic diversity observed in individuals affected with HF in Gran Canaria.

## 4. Materials and Methods

### 4.1. Subjects

The study population included families attending the Lipids Unit of the Complejo Hospitalario Universitario Insular Materno-Infantil de Gran Canaria. This cohort received a genetic diagnosis of FH, carried the p.(Tyr400_Phe402del) variant in *LDLR* and had both parents born on the island. We selected 11 unrelated family trios of p.(Tyr400_Phe402del)-mutation carriers and a homozygous individual. The trios were either mother–father–proband, or parent–proband–sibling.

In addition, 48 unrelated Canary Islanders not bearing the p.(Tyr400_Phe402del) mutation, who self-declared as having two generations of ancestors born in the Canary Islands, were included as controls.

### 4.2. Microsatellite Genotyping

Genomic DNA was extracted from whole blood samples preserved in EDTA using a salt precipitation protocol [34]. Fourteen microsatellite markers covering 5.4 Mbp (8.86 cM) flanking the p.(Tyr400_Phe402del) mutation (Table 2 and Figure 4) were genotyped in the cases and controls.

Amplifications were carried out in 10 μL volume PCRs containing 1× colourless GoTaq^®^ Flexi Buffer (Promega, Madison, WI, USA), 1.5 mm of MgCl_2_, 0.2 mm of each dNTP, 0.12 mm of each primer (see Table 2), and 0.1 U of *Taq* polymerase (Promega). The PCR programme consisted of 95 °C for 3 min, followed by 28 cycles (95 °C for 30 s, 58 °C for 15 s and 72 °C for 1 min) with a final extension at 72 °C for 10 min. Fluorescently labelled fragments were run on an ABI PRISM 3100 DNA sequencer (Applied Biosystems, Foster City, CA, USA) with the GeneScan-500 (LIZ) size standard. Alleles were scored using Peak Scanner™ Software v1.0 (Applied Biosystems).

### 4.3. Genetic Characterisation

Measures of genetic diversity, such as the total number of alleles per locus, mean observed (H_O_) and mean expected (H_E_) heterozygosities, were calculated using ARLEQUIN version 3.5.2.2 [35]. The same resource was used to test for departures from the Hardy–Weinberg equilibrium (HWE) and deviations from the linkage equilibrium (LD) for all pairwise locus combinations. A sequential Bonferroni correction [36] was applied to the HWE and LD results.

### 4.4. Estimation of the Age of the Variant

To estimate the age of the p.(Tyr400_Phe402del) mutation we used the Gamma linkage disequilibrium method (with correlated genealogy) implemented in the R Shiny app Genetic Mutation Age Estimator (https://shiny.wehi.edu.au/rafehi.h/mutation-dating/ (accessed on 8 May 2023)), which is fully described by Gandolfo et al. in 2014 [22]. This method estimates the age of a genetic mutation based on the genetic length of ancestral haplotypes common to individuals who share the mutation. Furthermore, this method has the advantage of using the information of the genomic distances and recombination rates of the microsatellite markers used for genotyping the study cohort. In this study, haplotypes were reconstructed based on genotypic information from relatives of mutation carriers.

### 4.5. Functional Characterisation of the Variant

#### 4.5.1. Cloning of LDLR Variant

A DNA fragment representing *LDLR* cDNA (NM_000527.4) containing the p.(Tyr400_Phe402del) variant was synthesized and cloned into the mammalian expression vector pcDNA3 (Genescript, Piscataway, NJ, USA). The resulting clones were Sanger-sequenced to verify accuracy.

#### 4.5.2. CHO-ldlA7 Cell Culture and Transfection

CHO-*ldl*A7 cells not expressing LDLr (kindly provided by Professor M. Krieger, MIT, MA, USA) were maintained in Ham’s F-12 medium containing 10% fetal bovine serum (FBS), 0.29 mg/mL of L-glutamine and antibiotics (0.75 mg/mL of penicillin; 100 μg/mL of streptomycin). Cells at 80% confluency were transfected with Lipofectamine^TM^ LTX using PLUS^TM^ Reagent (Invitrogen) following the manufacturer’s recommendations. LDLr functionality was assessed 48 h after transfection.

#### 4.5.3. Immunodetection of LDLr by Western Blot

Cells were lysed in ice cold 50 mM of Tris-HCl buffer containing 125 mM of NaCl, 1% Nonidet P-40, 5.3 mM of NaF, 1.5 mM of Na_4_P_2_O_7_ decahydrate, 1 mM of orthovanadate, 1 mg/mL of complete EDTA-free protease inhibitor cocktail (Roche, Basel, Switzerland), 0.25 mg/mL of Pefabloc and 4-(2-aminoethyl)-benzenesulfonyl fluoride hydrochloride (AEBSF; Roche), at pH 7.5. Cells were rotated at 4 °C for an hour, sonicated and centrifuged at 12,000× *g* for 15 min to remove insoluble material. Proteins were resolved by electrophoresis on nonreducing 8.5% SDS-PAGE and transferred to a nitrocellulose membrane for LDLR detection. Rabbit polyclonal anti-LDLR antibody (1:500) (Progen Biotechnik GimbH, Heidelberg, Germany) and mouse monoclonal anti-GAPDH antibody (1:1000) (Nordic Biosite, Little Chalfont, UK) primary antibodies were incubated overnight at 4 °C, while IRDye 680RD Goat anti-Mouse IgG and IRDye 800CW Donkey anti-Rabbit IgG (LI-COR) secondary antibodies were incubated at room temperature for 1 h.

Signals were developed using SuperSignal West Dura Extended Substrate (Pierce Biotechnology, Rockford, IL, USA) in a ChemiDoc XRS (Bio-Rad, Hercules, CA, USA).

#### 4.5.4. Analysis of LDLR Expression by Fluorescent Activated Cell Sorter (FACS)

LDLr expression at the cell membrane was assessed in a CytoFLEX Flow Cytometer (Beckman Coulter, Brea, CA, USA) using a mouse monoclonal antihuman-LDLR (C7) (1:100; 2.5 mg/L; Origene, Rockville, MD, USA) and an Alexa Fluor 488-conjugated goat antimouse IgG (1:200; Molecular Probes, Eugene, OR, USA) as primary and secondary antibodies, respectively, as previously described [37]. Each sample was performed in triplicate, and 10,000 events were acquired for data analysis.

#### 4.5.5. Analysis of LDL Uptake by FACS

Forty-eight hours post-transfection, cells were incubated with FITC-LDL (20 μg/mL) for 4 h at 37 °C to determine LDL uptake, as previously described [37]. For determining LDLR expression, cells were washed out with PBS-1% BSA, fixed in 4% paraformaldehyde for 10 min at room temperature and washed again to remove residual fixative. To determine the amount of internalized LDL, Trypan blue solution (Sigma-Aldrich, Steinheim, Germany) was added directly to the samples to a final concentration of 0.2%. Each sample was performed in triplicate, and 10,000 events were acquired for data analysis.

#### 4.5.6. Confocal Laser Scanning Microscopy

Confocal laser scanning microscopy was used to analyse LDLR expression and colocalization with the endoplasmic reticulum (ER)-specific marker calregulin. Cells transfected with the *LDLR-containing* plasmids were cultured for 48 h at 37 °C in 5% CO_2_. Then, the cells were washed twice with PBS-1% BSA, fixed with 4% paraformaldehyde for 10 min, washed and permeabilised with 1% TritonX-100 for 30 min at room temperature. Samples were blocked in PBS-10% FBS for 1h and incubated with the appropriate primary antibodies for 16 h at 4 °C, followed by incubation with the appropriate fluorescent secondary antibodies. Coverslips were mounted on a glass slide, and samples were visualised using a confocal microscope (Olympus IX 81, Tokyo, Japan) with sequential excitation and capture image acquisition with a digital camera (Axiocam NRc5; Zeiss, Jena, Germany). Images were processed using Fluoview v50 software (Olympus, Miami, FL, USA).

#### 4.5.7. Statistical Analysis

All measurements were performed at least 3 times unless otherwise specified, and results represent the mean ± standard deviation (SD). The differences between *LDLR* variants and wild-type (wt) *LDLR* were tested by a two-tailed Student’s *t*-test with a significance level of 0.05.

## 5. Conclusions

The evidence presented in this study suggests that the most prevalent mutation causing HF in the population of Gran Canaria, p.(Tyr400_Phe402del) in *LDLR*, was introduced or arose in the population after the Spanish colonisation of the Canarian Archipelago, which took place during the 15th century. This relatively recent mutation expresses a misfolded protein that is retained in the ER, preventing its expression at the cellular surface. Therefore, this in-frame deletion can be classified as a class 2a, defective, pathogenic *LDLR* variant.

## Figures and Tables

**Figure 2 ijms-24-11319-f002:**
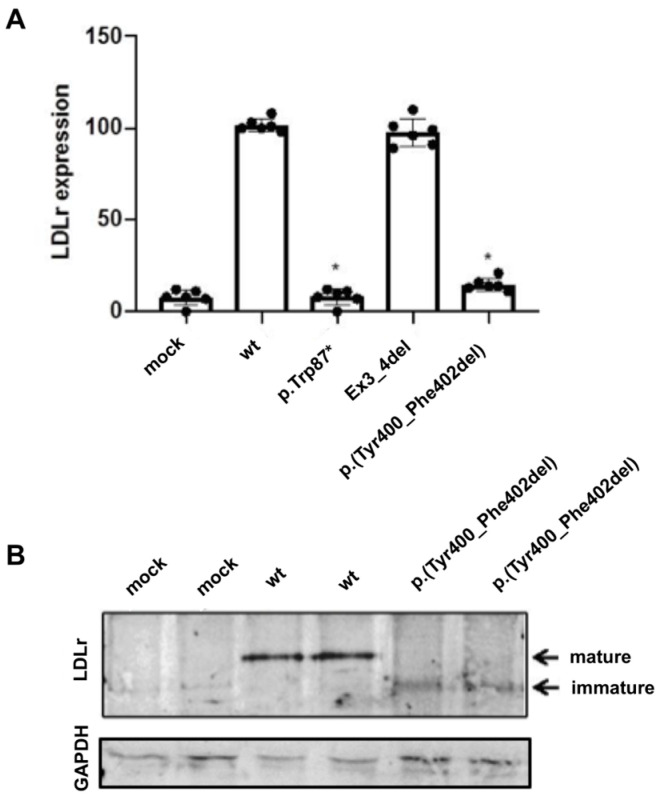
Expression of the p.(Tyr400_Phe402del) LDLR variant determined by (**A**) flow cytometry and (**B**) Western blot. Expression of LDLR variants was assessed in CHO-*ldlA7*-transfected cells as described in Section 4 (Materials and Methods). LDLR expression was assessed, by flow cytometry and Western blot 48 h post-transfection with the plasmids carrying the different *LDLR* variants. The values in A represents the mean of triplicates (*n* = 3). A representative blot is shown in panel B. Error bars in A represent ± SD. * *p* < 0.001 compared to wt using Student’s *t*-test.

**Figure 3 ijms-24-11319-f003:**
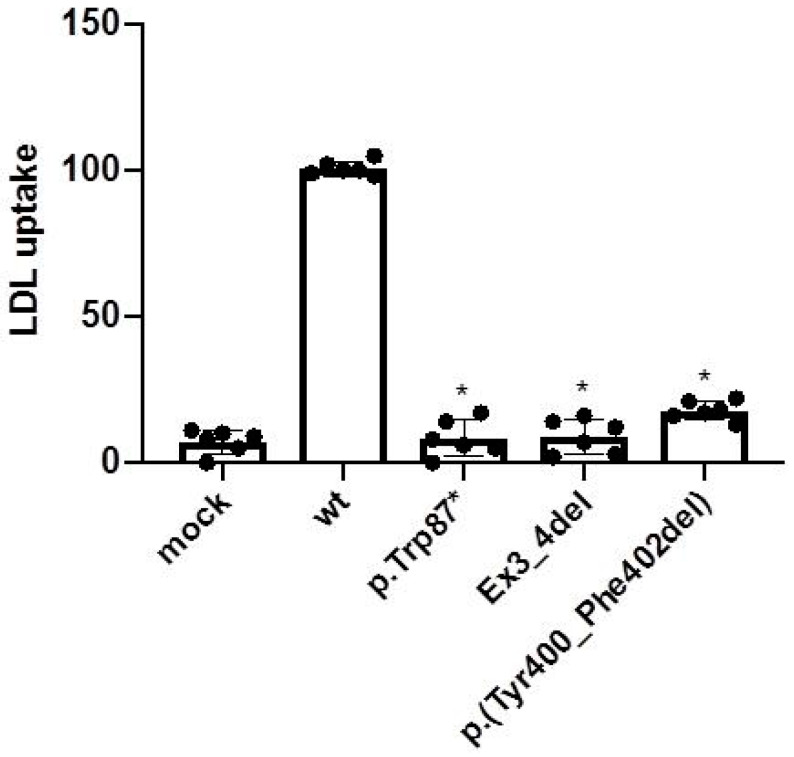
LDL uptake by the p.(Tyr400_Phe402del) LDLR variant determined by flow cytometry. LDL uptake by the LDLR variants was assessed in CHO-*ldlA7*-transfected cells as described in Section 4 (Materials and Methods). LDL uptake was assessed, using flow cytometry, 48 h post-transfection with the plasmids carrying the different *LDLR* variants. The values represent the mean of triplicates (*n* = 3) ± SD. * *p* < 0.001 compared to wt using Student’s *t*-test.

**Figure 4 ijms-24-11319-f004:**
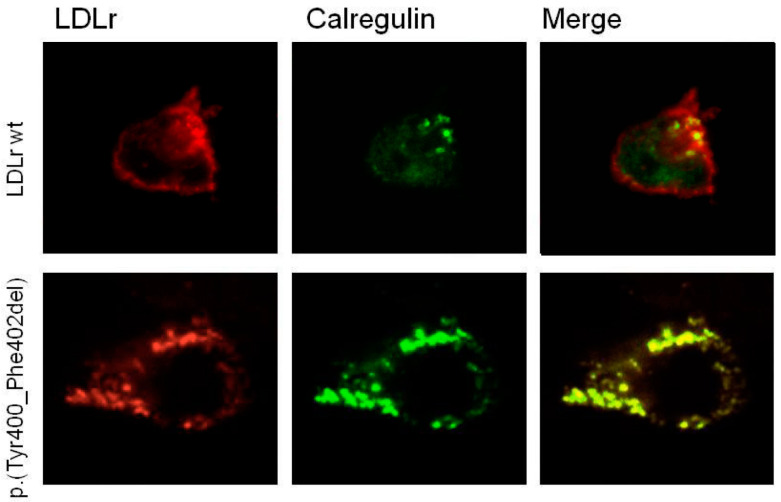
The p.(Tyr400_Phe402del) LDLR variant colocalised with calregulin in the ER. Confocal analysis of LDLR colocalisation with the ER was performed in wt and p.(Tyr400_Phe402del) LDLR variant with the ER-specific marker calregulin. Transfected cells were immunostained, as described in Section 4 (Materials and Methods). The images show a representative individual cell (magnification 60×).

**Table 1 ijms-24-11319-t001:** Demographic and clinical information of p.(Tyr400_Phe402del) carriers.

Gender (% females)	42.3
Age (years)	45.1 ± 18.4
BMI (Kg/m^2^)	25.5 ± 5.2
Tendinous xanthomas (%)	42.3
ASCVD (%)	15.4
Total cholesterol (mg/dL)	385.1 ± 69.2
HDL-c (mg/dL)	62.2 ± 20.3
LDL-c (mg/dL)	299.2 ± 60.7
Triglycerides (mg/dL)	100.2 ± 42
Lipoprotein (a) (mg/dL), median [IQR]	30 [13.5–85]

Values correspond to average ± standard deviation unless otherwise specified. BMI, body mass index; ASCVD, atherosclerotic cardiovascular disease; HDL-c, high-density lipoprotein cholesterol; LDL-c, low-density lipoprotein cholesterol; IQR, interquartile range.

**Table 2 ijms-24-11319-t002:** Primer sequences, description, and characterisation of the fourteen microsatellite loci analysed.

Locus.	Primer Sequence (5′-3′) Forward	Primer Sequence (5′-3′) Reverse	Label ^a^	Repeat Motif	Genomic Location ^‡^		N_A_	H_O_ *	H_E_ *
L10	**GAGGCTGAGACGGGAGAATC**	TTCCCCAACACACAAAGCAG	6-FAM	CA	8,284,611	8,284,940	13	0.833	0.825
L9	**CATGCTCAGCTTCCCAAGAC**	AGGTGGAGGTTGCAGTGAG	PET	GT	8,518,249	8,518,442	6	0.712	0.680
L8	**GACTTAGAATGGTGCCTGGC**	AAAATTAGCTGGGCACGGTG	NED	GT	8,622,652	8,622,894	9	0.551	0.592
L7	**GTTTCTCACGGCTGACTTGG**	CACCTGGCCTCACTTGATGT	VIC	AG	8,694,731	8,695,961	12	0.897	0.889
L6	**GGATGAGTGTGCTTTCTACCC**	GGCCCCATATGAACCGTTTC	6-FAM	GT	8,928,205	8,928,445	7	0.645	0.745
L5	**GCTATTTGGGGTCTCTATCAATG**	GAAATCGCACAGTATTTGTCTCAC	VIC	CA	9,067,697	9,067,918	13	0.667	0.821
L4	**AGAAGCTAGGACCACAGACG**	ATGCACACCTGTAGTCCCAG	NED	TG	9,501,287	9,501,509	10	0.854	0.821
L3	**GGGTCTGAGGATGTTTCTGC**	GCAAATATCCACTGCCCTTG	NED	GT	10,451,997	10,452,137	9	0.658	0.670
L2	**GGGTGCTAGGATTTGGGACT**	CATTTGGTCTTGCTCCTCTGA	PET	GT	10,794,316	10,794,475	9	0.444	0.445
L1	**AGTGTGGAAGGAAAAGGGAC**	CCAATTCTAGATGGGTCG	6-FAM	ATA	11,092,150	11,092,197	6	0.256	0.246
R1	**TCCAGCAATTGTTCCCATTCTC**	TACACAAACATTAGCCGGGC	6-FAM	TA	11,609,282	11,609,694	16	0.274	0.646
R2	**AGATCGCACCACTGTACTCC**	TTCCCGCCTAGTAACGGAC	VIC	CA	11,815,227	11,815,384	16	0.793	0.829
R3	**TCTTCCCATTGCAGTTGTGG**	AACACCCTCCCCATGTACAC	PET	GT	12,988,231	12,988,486	9	0.778	0.708
R4	ATAGGCCAAGACTGTCTAAAACAA	**GCCCTAACTGCTGTAAGAGAACT**	6-FAM	CA	13,730,141	13,730,344	9	0.737	0.683

^a^ Labelled primers are depicted in bold font. ^‡^ Genomic location based on Human Genome Assembly (GRCh38.p13). N_A_, number of alleles. H_O_, observed heterozygosity. H_E_, expected heterozygosity. * Values obtained from controls. Loci departing from Hardy–Weinberg equilibrium (*p* < 0.05) are underlined.

## Data Availability

Data is provided as Appendix A.

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
