# Peer review of "Age, Origin and Functional Study of the Prevalent LDLR Mutation Causing Familial Hypercholesterolaemia in Gran Canaria"

_ijms, 2023, doi:10.3390/ijms241411319_

Round 1

Reviewer 1 Report

It was a pleasure reviewing the manuscript of Suárez et al. study. This is a comprehensive study of the prevalent p.(Tyr400_Phe402del) mutation of the LDL receptor gene. The strength of the study includes genetic characterisation with the age of the variant estimation, in combination with the detailed functional variant analysis.

The manuscript is well written, the content is concise and understandable. The work is interesting and worthy of publication. I include minor notes/comments below.

Introduction:

Line 42- the author refers to publications from 2015 and 2016 stating a prevalence of the homozygous form of FH of 1:300,000 to 1:450,000. There are more recent reports in the literature indicating a prevalence of- 1:250,000 to 1:360,000 (e.g. Marina Cuchel et al., 2023 Update on European Atherosclerosis Society Consensus Statement on Homozygous Familial Hypercholesterolaemia: new treatments and clinical guidance, European Heart Journal, 2023; https://doi.org/10.1093/eurheartj/ehad197). With regard to the prevalence of the heterozygous form- the prevalence given is still recognised, although perhaps the source of the information could be more up-to-date (2013 article cited).

Materials and methods:

The description of the study group is quite limited, lacking basic information on phenotype or DLCN score.

For the analysis of population genetics, the ARLEQUIN tool was used, which is a free, frequently cited and well-regarded programme for the study of population structure, genetic migration, natural selection or genetic diversity. A less obvious choice seems to be the use of a web application to estimate the age of a genetic variant. Typically, tools such as BEAST, MEGA or PAML are chosen for such analysis.  Could the authors justify the choice of this rather unpopular tool?

Overall, I highly recommend the manuscript for publication.

Author Response

We would like to thank the reviewers for their positive comments and suggestions (in black font) that help us to improve the submitted version of our manuscript. Here we summarise our responses (in blue font) and the most significant changes are referred to by the line numbers in the revised manuscript.

We have also carefully reviewed the text for errors in expression and spelling.

We hope that these replies and changes to the manuscript appropriately address the reviewers’ comments and that the paper is now acceptable for publication.

Reviewer 1

It was a pleasure reviewing the manuscript of Suárez et al. study. This is a comprehensive study of the prevalent p.(Tyr400_Phe402del) mutation of the LDL receptor gene. The strength of the study includes genetic characterisation with the age of the variant estimation, in combination with the detailed functional variant analysis.

The manuscript is well written, the content is concise and understandable. The work is interesting and worthy of publication. I include minor notes/comments below.

Thanks for your encouraging comments.

Introduction:

Line 42- the author refers to publications from 2015 and 2016 stating a prevalence of the homozygous form of FH of 1:300,000 to 1:450,000. There are more recent reports in the literature indicating a prevalence of- 1:250,000 to 1:360,000 (e.g. Marina Cuchel et al., 2023 Update on European Atherosclerosis Society Consensus Statement on Homozygous Familial Hypercholesterolaemia: new treatments and clinical guidance, European Heart Journal, 2023; https://doi.org/10.1093/eurheartj/ehad197). With regard to the prevalence of the heterozygous form- the prevalence given is still recognised, although perhaps the source of the information could be more up-to-date (2013 article cited).

References have been updated (lines 41-42).

Materials and methods:

The description of the study group is quite limited, lacking basic information on phenotype or DLCN score.

We have included a new Table (Table 2 - line 210) with demographic and clinical information of the study group.

For the analysis of population genetics, the ARLEQUIN tool was used, which is a free, frequently cited and well-regarded programme for the study of population structure, genetic migration, natural selection or genetic diversity. A less obvious choice seems to be the use of a web application to estimate the age of a genetic variant. Typically, tools such as BEAST, MEGA or PAML are chosen for such analysis.  Could the authors justify the choice of this rather unpopular tool?

We agree with Reviewer 1 that other tools, such as BEAST, are more popular and offer the possibility to address many other evolutionary questions. However, we opted to use the less popular “Genetic Mutation Age Estimation” tool, as it has been specifically developed to estimate the age of particular mutations, which was the main purpose in our study. In addition, this tool has been developed by experts in the field of Genetics and has been used by many authors, as demonstrated by its citation in other prestigious journals such as Nature Immunology (Park et al 2020, Nat Immunol21:857-67), Nature Communications (Florian et al 2019, Nat Commun 10:4919), and Brain (Alistair et al 2021, Brain 144:584-600). In this regard, we have included additional information on this tool in lines 127 – 134.

Overall, I highly recommend the manuscript for publication. 

Reviewer 2 Report

I have read with great interest the article entitled "Age, origin and functional study of the prevalent LDLR muta- 2 tion causing familial hypercholesterolaemia in Gran Canaria". Overall, very interesting and meaningful. Several advice might help improve the manuscript:

1. The authors determined the age of the p.(Tyr400_Phe402del) mutation in the second part of the Results. Please specify the detailed analysis methods.

2. The calregulin punctua of the LDLr wt group in Figure 4 was not persuasive to prove the colocolization between LDLr variant and ER. Please provide more convincing images.

3. Please check and modify all figure legends in the manuscript appropriately.

The language of this manuscript is easy to understad.

Author Response

We would like to thank the reviewers for their positive comments and suggestions (in black font) that help us to improve the submitted version of our manuscript. Here we summarise our responses (in blue font) and the most significant changes are referred to by the line numbers in the revised manuscript.

We have also carefully reviewed the text for errors in expression and spelling.

We hope that these replies and changes to the manuscript appropriately address the reviewers’ comments and that the paper is now acceptable for publication.

Reviewer 2

I have read with great interest the article entitled "Age, origin and functional study of the prevalent LDLR mutation causing familial hypercholesterolaemia in Gran Canaria". Overall, very interesting and meaningful.

Thanks for the positive comments.

 Several advice might help improve the manuscript:

  1. The authors determined the age of the p.(Tyr400_Phe402del) mutation in the second part of the Results. Please specify the detailed analysis methods.

We have included further details of the applied method in the manuscript (lines 127 – 134). However, the full description of this method can be obtained from Gandolfo et al 2014 (cited in the manuscript).

  1. The calregulin punctua of the LDLr wt group in Figure 4 was not persuasive to prove the colocolization between LDLr variant and ER. Please provide more convincing images.

We have now provided a new Figure 4, which hopefully clarifies the colocalisation of the LDLr variant and the ER.

  1. Please check and modify all figure legends in the manuscript appropriately.

We have checked and modified some of the legends to clarify the corresponding figures.

Round 2

Reviewer 2 Report

Thanks for the authors' positive responses and all issues have been well solved. Therefore, this article is recommended for publication in IJMS.